# Mild COVID-19 in an APECED Patient with Chronic Inflammatory Demyelinating Polyneuropathy (CIDP) and High Titer of Type 1 IFN-Abs: A Case Report

**DOI:** 10.3390/pathogens12030403

**Published:** 2023-03-02

**Authors:** Mariella Valenzise, Simone Foti Randazzese, Fabio Toscano, Fortunato Lombardo, Giuseppina Salzano, Cristina Pajno, Malgorzata Wasniewska, Antonio Cascio, Maureen A Su

**Affiliations:** 1Department of Human Pathology of Adulthood and Childhood, University of Messina, 98121 Messina, Italy; 2Department Maternal and Child Health, Urological Sciences-Sapienza University, 00161 Rome, Italy; 3Department of Health Promotion, Maternal and Infant Care, Internal Medicine and Medical Specialties, University of Palermo, 90100 Palermo, Italy; 4Infectious and Tropical Diseases Unit, AOU Policlinico “P. Giaccone”, 90100 Palermo, Italy; 5Department of Microbiology, Immunology, Molecular Genetics University of California, Los Angeles, CA 90095, USA

**Keywords:** APECED, type 1 IFN-Abs, COVID-19, corticosteroids, subcutaneous immunoglobulins, case report

## Abstract

Autoimmune-Poly-Endocrinopathy-Candidiasis-Ectodermal Dystrophy (APECED), caused by mutations in the Autoimmune Regulator (AIRE) gene, is an autosomal recessive multi-organ autoimmunity syndrome usually defined by high serum titers of type I Interferon Autoantibodies (Type 1 IFN-Abs). These antibodies have recently been found in individuals in the general population who develop life-threatening Coronavirus Disease 2019 (COVID-19), but the significance of pre-existing Type 1 IFN-Abs in APECED patients with COVID-19 remains unclear. Previous reports of COVID-19 outcomes in APECED patients have been divergent, and protective roles have been proposed for female sex, age <26 years, and immunomodulatory medications including intravenous immunoglobulin (IVIg). We report the case of a 30-year-old male APECED patient who experienced a Severe Acute Respiratory Syndrome Coronavirus 2 (SARS-CoV-2) infection with mild symptoms of fatigue and headache without respiratory distress and did not require hospitalization. He received a stress dose of hydrocortisone for adrenal insufficiency and continued on his baseline medications, including subcutaneous administration of Immunoglobulins (SCIgs) for chronic inflammatory demyelinating polyneuropathy (CIDP). Mild COVID-19 in a 30-year-old male patient with APECED and pre-existing Type 1 IFN-Abs was unexpected. Younger age and management of autoimmunity may have played a role.

## 1. Introduction

The APECED, also called Autoimmune Polyendocrine Syndrome Type 1 (APS-1) or Multiple Autoimmune Syndrome Type 1 (MAS-1), is a monogenic autoimmune disorder typically caused by biallelic deleterious mutations in the Autoimmune Regulator (AIRE) gene, mainly characterized by three main features: chronic mucocutaneous candidiasis (CMC), chronic hypoparathyroidism (CH) and Addison’s disease (AD) [1]. Secondary manifestations include a wide range of autoimmune conditions including alopecia, vitiligo, type 1 diabetes, and chronic inflammatory demyelinating polyneuropathy (CIDP). Type I Interferon Autoantibodies (Type 1 IFN-Abs) are present in >95% of APECED patients and are considered a diagnostic marker when one of the main or secondary clinical manifestations is present [1,2].

In APECED patients, mutations in the AIRE gene lead to defective T-cell tolerance associated with thymic escape of autoreactive T cells and development of a broad range of autoantibodies, including anticytokine antibodies such as Type 1 IFN-Abs [2]. These autoantibodies can also occur in other conditions; for example, during exogenous Type 1 IFN treatment or in women with Systemic Lupus Erythematosus (SLE) [3]. Recently, several studies have reported an association between these autoantibodies and severe Coronavirus Disease 2019 (COVID-19) in the general population [3,4,5], suggesting a pathogenic role for Type 1 IFN-Abs. Given this, it has been hypothesized that pre-existing high levels of Type 1 IFN-Abs may strongly predispose people with APECED to develop severe COVID-19 disease [3,4,5]. In support of this, three patients with APECED and severe COVID-19 were described in an early report [6]. Furthermore, in another case series, 15 of 22 APECED patients with COVID-19 required intensive care treatment, suggesting that severe illness occurs in the majority of APECED patients [4].

However, some case series of mild COVID-19 in APECED patients with high levels of Type 1 IFN-Abs have also been reported [6]. In a series of six APECED patients, four patients infected with COVID-19 had only mild symptoms, suggesting that APECED may not be a risk factor for severe COVID-19. All patients were females and younger than 26 years of age in this series, raising the possibility that sex and age may also play important roles [6]. Here, we describe the case of a 30-year-old male patient with APECED and pre-existing high titers of IFN-Abs who contracted Severe Acute Respiratory Syndrome Coronavirus 2 (SARS-CoV-2), developing mild COVID-19. The aim of our report is to underline possible protective factors associated with mild COVID-19 in patients with pre-existing IFN-Abs.

## 2. Case Presentation

D. is a 30-year-old male with APECED diagnosed at 1 year of age with compound heterozygous mutations in the AIRE gene. His family history was remarkable for APECED in an older brother, who died of chronic lung disease at 18 years of age [7,8]. As expected, given the diagnosis of APECED, high titers of autoantibodies—again, type 1 IFNs (IFN-Abs)—were detected. Since his first year of life, D. presented with recurrent bouts of oral candidiasis which was treated with itraconazole. He developed CH, which was treated with Calcium and Vitamin D, and AD, which was treated with Hydrocortisone (10 mg/m^2^/day), Fludrocortisone (0.1 mg/day). In addition, the patient developed vitiligo, alopecia, herpetic genital infections, and malabsorption. At the age of 15 years, he also developed chronic inflammatory demyelinating polyneuropathy (CIDP), the first APECED-related case described in the literature [9], for which he was treated with SubCutaneous Immunoglobulins (SCIgs). The patient was followed up via telemedicine during the COVID-19 pandemic. In April 2021, he was diagnosed with asymptomatic COVID-19: a SARS-CoV-2 Real Time-PCR test was performed because of a close contact with a SARS-CoV-2 infected relative and showed positive results. The patient was not vaccinated. In 3 days, he developed fatigue and mild headache. No dyspnea, cough, fever, or change in smell and taste were noted. He started prophylactic therapy with heparin, antibiotic (piperacillin/tazobactam), and stress dose hydrocortisone, and continued the SCIgs for CIDP. In 19 days, the infection was eradicated, confirmed by a SARS-CoV-2 Real Time-PCR negative test. The laboratory exams performed during the infection were normal (Table 1).

## 3. Discussion

The outbreak of COVID-19 in 2019 presented significant challenges worldwide. The anti-SARS-CoV-2 immune response has now been extensively studied, and defects in antiviral mechanisms, such as in IFN-I immunity, have been linked to disease severity [5]. Type 1 IFN response is crucial in protecting against viral diseases: it is important in limiting the early viremic phase in the first few days of the infection, as well as preventing the secondary phase of pulmonary and systemic hyperinflammation. Since the recent report of three unrelated APECED patients with high titers of Type 1 IFN-Abs who developed life-threatening COVID-19 pneumonia [6], it has been suggested that these neutralizing autoantibodies were linked to life-threatening manifestations. Additionally, at least 10% of 987 patients in the general population with life-threatening COVID-19 presented with high titers of Type 1 IFN-Abs, suggesting that these neutralizing autoantibodies contribute to poor outcome even in patients without APECED [3]. This study was followed by a report of 22 APECED patients aged 8 to 48 years old diagnosed with COVID-19 [4], 15 of whom (68%) were critically ill with hypoxemia, requiring admission to the Intensive Care Unit (ICU). Four patients (18%) died because of sepsis or respiratory failure, while the other 11 (82%) survived, but required intubation and mechanical ventilation. One of them required ExtraCorporeal Membrane Oxygenation (ECMO), two of them developed pneumothorax, and one of them was discharged with a tracheostomy. Together, these data suggest that neutralizing Type 1 IFN-Abs may lead to life-threatening COVID symptoms in infected individuals. Importantly, however, there is also a case series in which APECED patients with high titers of Type 1 IFN-Abs developed mild COVID-19 disease. Meisel et al. [6] suggested that young age (<26 years) and female sex may be protective. Concerning our case, the patient presented an asymptomatic disease, only developing mild manifestations, such as asthenia or headache. IVIg may play a role in rescue therapy for critical COVID-19 cases, so we supposed that chronic therapy with SCIgs for CIPD may have played a possible protective role in the clinical course of our APECED patient [4,10,11,12,13]. Bastard et al. in the previous series of 22 patients hypothesized possible protective factors for the 7 patients (34%) who did not become critically ill [4]. Interestingly, three patients were receiving monthly IVIg therapy at the time of the infection and another patient was hospitalized prophylactically and treated with subcutaneous recombinant IFN-beta and convalescent plasma therapy [4]. It could be possible that therapies with IVIg, SCIg, or plasma may decrease the pathogenicity of the autoantibodies against IFN, or they could act through other mechanisms [10,11,12,13,14]. Another hypothesis could be found in the early high dose of corticosteroids treatment, suggesting that it might prevent or attenuate the secondary hyper-inflammatory phase of the disease (Table 2) or prevent adrenal crisis in patient with pre-existing primary adrenal insufficiency.

Corticosteroids can reduce capillary dilation, inflammatory cell exudation, leukocyte infiltration, and phagocytosis in the early phase of inflammation, and can also inhibit the excessive proliferation of capillaries and fibroblasts in the late stage. However, treatment with corticosteroids is debated in COVID-patients and the most effective type of corticosteroid treatment is another concern that needs to be addressed. Therefore, there is an urgent need for more RCTs to confirm and validate previous findings [15].

Prevention also plays an important role for these patients; APECED patients should be prioritized for vaccination against COVID-19, and measures should be taken to avoid infection [4]. Telemedicine can be an important tool to help manage patients with high-risk underlying disease, by avoiding exposures during in-person follow-up [10]. Prompt treatment with available antiviral therapies is also an important strategy to prevent the progression to hypoxemic COVID-19 [4].

## 4. Conclusions

Mild COVID-19 in a 30-year-old male patient with APECED and pre-existing Type 1 IFN-Abs was unexpected. We do not exclude that younger age and management of autoimmunity may have contributed.

## Figures and Tables

**Table 1 pathogens-12-00403-t001:** Laboratory investigations.

Investigation	Patient’s Values	Laboratory Normal Values
Hemoglobin (Hb)	15.20 (g/dL)	13–17 (g/dL)
White Blood Cells (WBC)	11,700 (K/uL)	4000–10,000 (K/uL)
Neutrophils (N)	61.7 (%)	37–80 (%)
Lymphocytes (L)	30 (%)	10–50 (%)
Platelets (PLT)	229,000 (K/uL)	140,000–450,000 (K/uL)
C-Reactive Protein (CRP)	0.08 (mg/dL)	0–0.5 (mg/dL)
Prothrombin Time (PT)	684 (%)	70–130 (%)
International Normalized Ratio (INR)	1.21	-
Activated Partial Thromboplastin Time (aPTT)	26.7 (s)	22–30 (s)
Fibrinogen	217 (mg/dL)	170–450 (mg/dL)
Brain Natriuretic Peptide (BNP)	10 (pg/mL)	0–100 (pg/mL)

**Table 2 pathogens-12-00403-t002:** Comparison between APECED patients who experienced SARS-CoV-2 infection reported in literature and our patient: female sex and previous immunosuppressive therapy could play a protective role against life-threatening COVID-19 pneumonia.

References	Number of Patients	Median Age (Years)	Sex	High Titers of IFN Abs	Severity Disease	Chronic Treatment for Mild Disease	Chronic Treatment for Life Threatening Disease	Outcome
[4]	22	24.5	F: 13 M: 9	Present: 21 Not tested: 1	Mild–moderate: 7 Life-threatening: 15 (M: 6, F: 9)	Hydrocortisone and/or fludrocortisone (*n* = 6); fluconazole (*n* = 3); rituximab and IVIgs (pz n 6), ruxolitinib (pz n 15), monthly IVIgs (pz n 4 and 10), supportive and other non-immunosuppressive therapy.	Hydrocortisone and/or fludrocortisone (*n* = 14); fluconazole (*n* = 5); tacrolimus (pz n 18), supportive and other non-immunosuppressive therapy.	Survival: 18 Death: 4 (sepsis and/or respiratory failure).
[6]	4	18.15	F: 4 M: 0	Present: 4	Mild: 4 Life-threatening: 0	Hydrocortisone and/or fludrocortisone (*n* = 2); supportive and other non-immunosuppressive therapy.	-	Survival: 4
Our Experience	1	30	M	Present	Mild	Hydrocortisone and fludrocortisone; itraconazole; SCIgs; supportive and other non-immunosuppressive therapy.	-	Survival

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
