# Peer review of "Mild COVID-19 in an APECED Patient with Chronic Inflammatory Demyelinating Polyneuropathy (CIDP) and High Titer of Type 1 IFN-Abs: A Case Report"

_pathogens, 2023, doi:10.3390/pathogens12030403_

Round 1
Reviewer 1 Report
The case is clearly written. However there is not much elaboration on how could he done well. Is it because of the IG? Better support should be offered as an explanation. The author s mention the use of steroids.. what would be the proposed mechanism?
Author Response
Dear Reviewer,
we are grateful for your comments.
In the discussion (lines 118-129) we underline that chronic treatment with SCIgs may have played a protective role against the development of severe SARS-COV-2 infection thanks to their role against the hyperinflammation, typical condition of severe COVID-19 disease, as reported in the paper of Bastard et al.
In our patient the use of corticosteroids in course of acute disease might prevent or attenuate the secondary hyper-inflammatory phase of the disease (lines 128-130 of the disccussion).
13.11.2022
We thank the referee and we agree with the comments regarding the limit of a description of a single clinical case.
We corrected the term “penetration” at 65 line and highlighted the corrections in yellow.
Aim of our study is not to emphasize the indiscriminate use of corticosteroids in all patients with covid 19, but to underline a possible and potential protective effect of the stress dose corticosteroids in APECED patients with SARS COV 2 infection.
Although the role of corticosteroids in blocking the inflammatory cascade of SARS Cov 2 infection is much debated in the international scientific literature, as several manuscripts confirm the negative role and few ones underline a protective role, the description of our case showed that the combination therapy of a “stress dose” of corticosteroids for adrenal insufficiency and intravenous immunoglobulins for chronic demyelinating polyneuropathy seems to have a protective effect in APECED patients with SARS Cov 2 infection avoiding a fatal evolution of COVID disease.
The potential protective role of corticosteroids reported in the description of our single clinical case underlined the importance of “stress dose “ of cortcosteroid in APECED-COVID patients in order to prevent adrenal crisis (as reported in the new version of the paper). In these patients with SARS Cov 2 infection a fatal clinical course would be expected because they have positive anti-inteferon omega antibodies, as expression of subverted immunity.

Reviewer 2 Report
It is an interesting observation that APECED patients diagnosed with COVID may benefit from corticosteroid and SCIgG treatment in order to reduce the risk of severe COVID.
Here are some more specific comments:
1. What is the main question addressed by the research? This paper is a case report describing the outcome of a COVID infection in an APECED patient with CIDP. Patients with APECED have elevated levels of type 1 IFN autoantibodies, and previous papers have described an association between APECED and severe COVID. Other papers have suggested that APECED patients generally have severe COVID. However, the patient described in this report also received corticosteroids and SCIgs as part of their treatment for chronic hypoparathyroidism and for chronic inflammatory demyelinating polyneuropathy (CIDP). When this patient developed COVID, his symptoms were mild in comparison with report COVID infection in APECED patients. The authors speculate that because this patient was receiving corticosteroids and SCIgs, that diminished the severity of the COVID. This might be a possible treatment for APECED patients that acquire COVID. 2. Do you consider the topic original or relevant in the field? Does itaddress a specific gap in the field? This topic is highly relevant as it offers an adjuvant treatment of COVID in APECED patients. Of course, this is only one patient described, but it should it something that other's treating APECED patients might want to keep in mind.
3. What does it add to the subject area compared with other published
material?
4. What specific improvements should the authors consider regarding the
methodology? What further controls should be considered? As this one a case study of one patient, it would be nice if the authors could find evidence of other APECED patients that had received corticosteroids and/or SCIgs that also had a mild case of COVID.
5. Are the conclusions consistent with the evidence and arguments presented
and do they address the main question posed? Yes
6. Are the references appropriate? Yes
7. Please include any additional comments on the tables and figures. Tables were fine.
Author Response
Dear Reviewer,
we are grateful for your comments.
We underline the role of the chronic treatment with SCIgs against the hyperinflammation in severe COVID-19 disease (lines 118-128 of the discussion): they may decrease the pathogenicity of the auto-antibodies against IFN or they could act through other mechanisms, confirming that the immunosuppressive therapy may play a role in the prevevention of severe SARS-COV-2 infections. Another hypothesis could be found in the early high dose of corticosteroids treatment (stress-dose), suggesting that it might prevent or attenuate the secondary hyper-inflammatory phase of the disease, if started promptly in hypoadrenal patients
13.01.2023
We thank the referee and we agree with the comments regarding the limit of a description of a single clinical case.
We corrected the term “penetration” at 65 line and highlighted the corrections in yellow.
Aim of our study is not to emphasize the indiscriminate use of corticosteroids in all patients with covid 19, but to underline a possible and potential protective effect of the stress dose corticosteroids in APECED patients with SARS COV 2 infection.
Although the role of corticosteroids in blocking the inflammatory cascade of SARS Cov 2 infection is much debated in the international scientific literature, as several manuscripts confirm the negative role and few ones underline a protective role, the description of our case showed that the combination therapy of a “stress dose” of corticosteroids for adrenal insufficiency and intravenous immunoglobulins for chronic demyelinating polyneuropathy seems to have a protective effect in APECED patients with SARS Cov 2 infection avoiding a fatal evolution of COVID disease.
The potential protective role of corticosteroids reported in the description of our single clinical case underlined the importance of “stress dose “ of cortcosteroid in APECED-COVID patients in order to prevent adrenal crisis (as reported in the new version of the paper). In these patients with SARS Cov 2 infection a fatal clinical course would be expected because they have positive anti-inteferon omega antibodies, as expression of subverted immunity.
